# The Bark Beetle *Phloeotribus rhododactylus* (Coleoptera: Curculionidae) Has a Stable Range in Europe

**DOI:** 10.3390/insects11120856

**Published:** 2020-12-02

**Authors:** Tomáš Fiala, Jaroslav Holuša

**Affiliations:** Faculty of Forestry and Wood Sciences, Czech University of Life Sciences Prague, Kamýcka 1176, 16500 Prague, Czech Republic; holusaj@seznam.cz

**Keywords:** bionomics, *Cytisus scoparius*, Czech Republic, occurrence, resource, Scolytinae

## Abstract

**Simple Summary:**

The abundance of bark beetles is generally explained by resource-related parameters. The bark beetle *Phloeotribus rhododactylus* feeds mainly on the shrub *Cytisus scoparius*. Other host plants include *Spartium junceum*, *Cytisus* sp., *Ulex europaeus*, *Calicotome* sp., *Coronilla emeroides*, *Genista florida*, *Adenocarpus complicatus*, and *Ficus carica*. *Phloeotribus rhododactylus* seems to have a stable range that is centred in Western Europe and extends to Eastern Europe. Its abundance is highest in Western Europe and decreases to the east, which coincides with the distribution of the host tree, *Cytisus scoparius*. Even though *Cytisus scoparius* is an invasive plant in agricultural and natural ecosystems out of Europe, *Phloeotribus rhododactylus* has not been found in any of the areas invaded by *Cytisus scoparius*.

**Abstract:**

The bark beetle *Phloeotribus rhododactylus* feeds mainly on the shrub *Cytisus scoparius.* The range of *P. rhododactylus* extends from Spain in the south to southern Sweden, Denmark, and Scotland in the north. Its range to the east extends to Poland, Slovakia, and Hungary, but single localities are known further east in Romania, Bulgaria, and Greece. It is clear that the range of the beetle matches that of its main host. *C. scoparius* is adapted to Mediterranean and coastal climates, and its range is limited by low winter temperatures. *P. rhododactylus* is, therefore, rare in Central Europe. It infests either individuals of *C. scoparius* that have been damaged by mammalian herbivores or snow or that are drought-stressed. Although *C. scoparius* is an invasive plant in agricultural and natural ecosystems, *P. rhododactylus* has not been found in any of the areas where *C. scoparius* has invaded.

## 1. Introduction

Scolytinae (Coleoptera: Curculionidae) includes both bark and ambrosia beetles and represents species of major economic and ecological importance in forests worldwide [1,2]. Most scolytine species feed on recently cut or injured tissues of woody plants, and such feeding can cause massive tree mortality depending on both tree health and beetle abundance [3,4,5].

Although weakened trees (i.e., wind-fallen, fire-injured, water-stressed, or trees damaged by other biotic factors) are highly attractive to Scolytinae, healthy trees are rarely attacked [6], and less than 1% of scolytine species regularly kill healthy standing trees [7]. Scolytinae affect forest dynamics by contributing to decomposition and nutrient cycling [3,8]. 

The absolute majority of Scolytinae beetles perforate the bark of trees and dig galleries near the cambium, but bark and ambrosia beetles differ in their feeding strategies. Bark beetles are mostly monophagous or oligophagous species that feed directly on phloem tissues (i.e., phloemophagy), whereas ambrosia beetles are polyphagous species that feed on fungi that they introduce and cultivate in their galleries (i.e., xylomycetophagy) and on xylem [6,9,10,11]. The lack of host specificity contributes to the invasiveness of ambrosia beetles in many forest ecosystems [12,13].

Scolytinae have been studied more than any other group of forest insects, but most investigations have been restricted to only a few pest species; see [14]. The abundance of scolytids is generally explained by resource-related parameters. In contrast to their abundance, the pest status of scolytids was previously found to be significantly related to species-specific traits, such as body size and maximum number of generations per year [14]. The latter study did not include the bark beetle *Phloeotribus rhododactylus* (Marsham, 1802) because information on its hosts was lacking [14].

*Phloeotribus rhododactylus* feeds mainly on the shrub *Cytisus scoparius* (L.) Link, 1822. Other host plants include *Spartium junceum* L., 1753; *Cytisus* sp., *Ulex europaeus* L., 1753; *Calicotome* sp.; *Coronilla emeroides* Boiss. & Spruner, 1843; *Genista florida* L., 1759; *Adenocarpus complicatus* (L.) J. Gay ex Gren. & Godr., 1848; and *Ficus carica* L., 1753 [15,16,17]. This bark beetle was previously reported to occur mainly in Western Europe and, to a lesser degree, in Central Europe, where its occurrence diminishes to the east [15]. Although *C. scoparius* is known to occur in most areas in Central Europe, *P. rhododactylus* has been considered rare in Central Europe [18,19]. 

The objectives of the current study were to summarize the occurrence of *P. rhododactylus* in Europe and to determine whether it remains rare in the Czech Republic. 

## 2. Bionomics of *Phloeotribus rhododactylus*

*Phloeotribus rhododactylus* feeds under the bark of the branches of *Cytisus scoparius*, *Spartium junceum*, and other shrubs. The larval corridors are long and sparse. It has only one generation per year. It flies in May, the larvae mature during the summer, and new beetles hatch at the end of the summer. The new beetles remain where they hatch for an extended period of maturation feeding and then overwinter [15]. The exit holes are round with diameters of 0.7 ± 0.2 mm (n = 40). The gallery system is in the shape of the letter Y with a width of 1.1 ± 0.1 mm and a length of 10–16 mm (n = 10); the larval corridors are 12–15 mm long (Figure 1; the photographs in this figure were taken with a TSC5005-S951B-V3 mobile camera with a resolution of 5.0 Mpx with a millimetre scale; measurements were made with the web application www.cnspg.cz/mince/index.php).

Little is known about the pheromone communication of *P. rhododactylus*. The spriroacetal, (E)-conophthorin, which has previously been reported from *Hylesinus varius* (Fabricius, 1775) [20] and *Taphrorychus bicolor* (Herbst, 1793) [21], was identified in the frass produced by boring beetles of *P. rhododactylus*. The antennae of *P. rhododactylus*, however, provided no clues about *P. rhododactylus* pheromones due to experimental difficulties with antennal preparation [22].

*Phloeotribus rhododactylus* may be a vector of the fungus *Geosmithia langdonii* (Kolařík, Kubátová and Pažoutová, 2005), although the latter authors studied only one specimen of the beetle [23].

*Geosmithia* sp. is a yeast and is found much less frequently in bark beetles in comparison with other yeasts or ophiostomatoid fungi [24]. *Geosmithia* sp. is also able to produce enzymes that can degrade lignocellulose-like substances [25]. The yeast converts food into an aggregation pheromone [26,27]. At the same time, a large population of yeasts in a weakened tree can increase its infestation by bark beetles [28]. In contrast, the defences of a healthy tree can reduce the diversity of yeasts that it supports [29].

Overall, yeasts are important to bark beetles; i.e., they affect beetle competitiveness and production of semiochemicals, serve as a food supplement, and influence the distribution of microorganisms that can affect the beetles in the subcortical environment [30,31]. Yeast development is affected by temperature. Yeast cannot survive at temperatures above 39 °C, and the optimum temperature range is 27–30 °C [32]. That temperature range is also the developmental optimum for bark beetles [33,34,35,36,37].

## 3. Common Broom *Cytisus scoparius*

*Cytisus scoparius*, the common broom or Scotch broom, is a perennial leguminous shrub native to all Europe, North Africa, and the Middle East [38]. Plants of *C. scoparius* typically grow to 1–3 m (3.3–9.8 ft) tall, rarely to 4 m (13 ft), with main stems up to 5 cm (2.0 in) thick, rarely 10 cm (3.9 in). The shrubs have green shoots with small deciduous trifoliate leaves 5–15 mm long, and in spring and summer, they are covered in profuse golden yellow flowers 20–30 mm from top to bottom and 15–20-mm wide.

This species is adapted to Mediterranean and coastal climates, and its range is limited by cold winter temperatures. Especially the seeds, seedlings, and young shoots are sensitive to frost, but adult plants are hardier, and branches affected by freezing temperatures regenerate quickly [39,40,41].

*C. scoparius* is found in sunny sites, usually on dry, sandy soils at low altitudes, tolerating very acidic soil conditions [39]. In some places outside of its native range, such as India, South America, western North America—particularly Vancouver Island, Washington, Oregon, and California, and west of the Cascade and Sierra Nevada mountains—Australia, and New Zealand (where it is a declared weed), it has become an ecologically destructive colonizing invasive species in grassland, shrub and woodland, and other habitats [42]. Over some parts of its native range, *C. scoparius* is an invasive plant in agricultural and natural ecosystems. In France, it is often considered an occupying species that preferentially colonizes abandoned pastures [43].

## 4. Geographic Range of *Phloeotribus rhododactylus*

The occurrence of *P. rhododactylus* has been already known in Western and Central Europe for a long time. In most countries, the first records date back to the 1950s, and only in few countries has it been reported later in the second part of the 20th century (Figure 2A,B), something that could be potentially associated with the low abundance of the insect. Even in one of the most comprehensive works on bark beetles based on extensive research by the author [44,45], the records of *P. rhododactylus* were mainly concentrated in Western Europe, being significantly rarer in the east [15].

Throughout the natural distribution of *P. rhododactylus*, we found records of its occurrence at 357 specific localities (Table 1). The number of localities is much greater for Western Europe (Great Britain, Germany, and France) than for Central or Eastern Europe (Figure 2A). 

The range of *P. rhododactylus* extends to Spain (probably northern Spain) in the south and to southern Sweden, Denmark, and Scotland in the north. The range to the east extends to Poland, Slovakia, and Hungary, but single localities are known further to the east in Romania, Bulgaria, and Greece (Figure 2A). It is likely that the small number of sites in south and east Europe is due to a lack of literary resources and not to the natural absence of beetles in these countries. Occurrences in Latvia and Macedonia have been reported but without designation of the specific locality [91]. 

Although *P. rhododactylus* occurs throughout Europe (Figure 2A) [91], it appears to be a Euro-Mediterranean species [92]. It is also widespread in non-European countries, such as Algeria, Egypt, Libya, Morocco, Madeira, Tunisia, and Iran [91]. Its abundance is highest Western Europe and decreases to the north and east. Its current distribution (Figure 2A) appears to be similar to that reported in 1955 [15].

The range of *P. rhododactylus* matches that of its main host, *C. scoparius*. *C. scoparius* is adapted to milder climates and its range is limited by cold winter temperatures. Although adult plants are relatively hardy, the seeds, seedlings, and young shoots of *C. scoparius* are sensitive to frost [39,40,41]. Its occurrence to the east (Russia and Ukraine) is sporadic [93].

*C. scoparius* has become an ecologically destructive invasive species in grasslands, shrublands, woodlands, and other habitats in India, South America, western North America, Australia, and New Zealand [42]. *P. rhododactylus* has not been found in any of the areas invaded by *C. scoparius* [3,94].

**Figure 2 insects-11-00856-f002:**
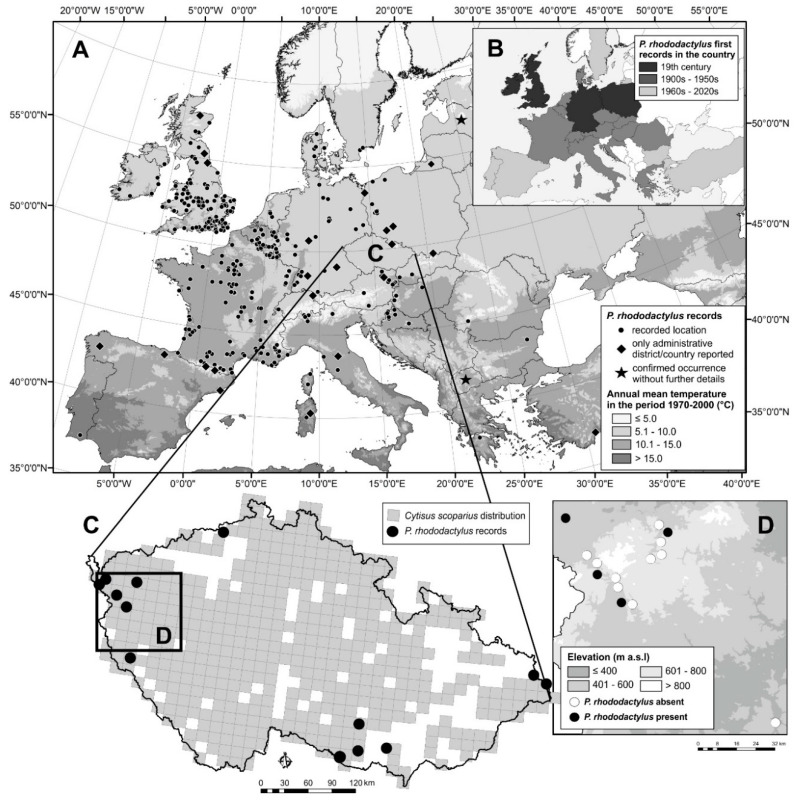
Localities (or countries and counties) where *Phloeotribus rhododactylus* has been reported to occur in Europe with annual mean temperature (**A**) (see Table 1); periods of its first reports in countries (**B**); occurrence of *Phloeotribus rhododactylus* with *Cytisus scoparius* in the Czech Republic (https://portal.nature.cz/nd/) (see Table 2) (**C**,**D**) in a recent study in western Bohemia, reported here (List of localities where *Phloeotribus rhododactylus* was not found in the study conducted by T. Fiala in 2020 in western Bohemia: Bečov na Teplou (50.0858942 N, 12.8625314 E); Blovice (49.5961411 N, 13.5222964 E); Dolní Žandov (50.0175272 N, 12.5845317 E); Dolní Žandov—Manský dvůr (50.0356972 N, 12.5319819 E); Kfely (50.1607864 N, 12.8421089 E); Mariánské Lázně (49.9555339 N, 12.6991647 E), NR Lazurový vrch (49.9135836 N, 12.7724494 E); NR Údolí Teplé (50.0554547 N, 12.8287328 E); Valy (49.9824786 N, 12.6813111 E); Vodná (50.1106978 N, 12.8631644 E)). NR—Natural Reserve.

## 5. Occurrence of *Phloeotribus rhododactylus* on the Eastern Edge of Its Range

In the last 60 years, *P. rhododactylus* has been detected in only 14 localities in the Czech Republic, with eight of them being after 2000 (Table 2). The localities lie at altitudes ranging from 290 to 1000 m a.s.l., but most of the localities are between 300 and 600 m a.s.l. (Figure 2C,D). An exception is the locality of the Velká Čantoryje Mt., which was deforested in the past, which allowed the occurrence of the Scotch broom and, subsequently, thanks to the connectivity of the landscape, also the bark beetle. In recent years, the species has been detected only in the western and south-eastern part of the country. *P. rhododactylus* has been known in the south-eastern Czech Republic since the 1920s [96], where the presence of a large population has been repeatedly confirmed (Table 2). We consider the localities in the northern and eastern part of the Czech Republic to be historical, because the species has not been recently reported from either area.

*C. scoparius* is abundant only in two areas where *P. rhododactylus* is known to occur (https://portal.nature.cz/publik_syst/nd_nalez-public.php?idTaxon=36486). Although *C. scoparius* is found almost everywhere, its distribution is uneven, and the sizes (areas) of the stands are not known. We suspect that large areas of *C. scoparius*, especially large areas that are near other large areas, have an increased probability of containing *P. rhododactylus*. The absence of large stands of *C. scoparius* in eastern Czech Republic probably explains the absence of *P. rhododactylus* in that part of the country (J. Holuša observ.).

In the western part of the country (Bohemia), in 2020, the first author inspected more than 10 weakened or dead *C. scoparius* branches in each of the 15 localities in which *C. scoparius* occupied more than 0.1 ha. *P. rhododactylus* occurrence was determined based on the presence of galleries (Figure 1A) and exit holes (Figure 1B). Rods were placed in emergence traps, and captured beetles were reared at 20 °C and light-dark 16-8 hrs. The beetles were identified to species level by the first author based on Pfeffer [15]. With these methods, *P. rhododactylus* was found in four localities (Table 2, Figure 2D) at altitudes ranging from 450 to 580 m a.s.l. They appeared consistently on shrubs that were damaged by game beating (90%) or broken by wet snow (10%). The beetles infested the lower third of trunks that were 1.5–2.5 cm in diameter. Exit holes numbered about five per m of branch, but thousands of entry holes per m of shrub were previously reported from Popice in south-eastern Czech Republic [95]. We suggest that this difference in *P. rhododactylus* abundance between the western and south-eastern regions of the Czech Republic can be explained by the drought and higher temperatures that occur in the south-eastern region [99]. Under severe water stress, trees may lack sufficient kino, resin, or latex to resist attack and are, therefore, more likely to be attacked by wood borers, secondary bark beetles, and latent pathogens [100,101].

## 6. Conclusions

*Phloeotribus rhododactylus* seems to have a stable range that is centred in Western Europe and extends to Eastern Europe. Its abundance is highest in Western Europe and decreases to the east, which coincides with the distribution of the host tree, *C. scoparius*. *P. rhododactylus* is a rare species in Central Europe. It occupies trees or shrubs that have been damaged or that are drought-stressed. It is possible that weak pheromone communication and weak interactions with fungi also contribute to its rare occurrence.

As is the case for abundance of most bark beetles [14], the abundance of *P. rhododactylus* can apparently be explained by resource-related parameters.

## Figures and Tables

**Figure 1 insects-11-00856-f001:**
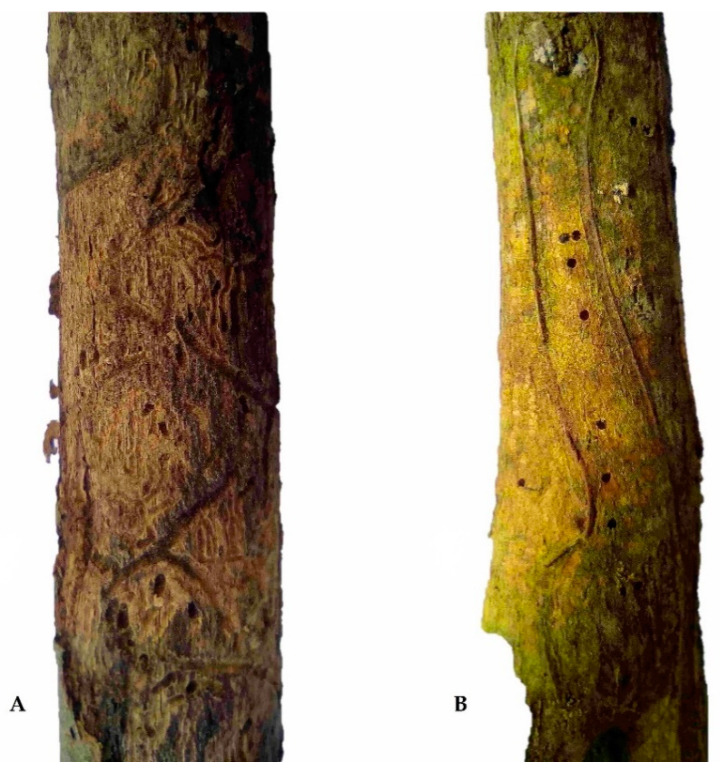
Galleries (**A**) and exit holes (**B**) of *Phloeotribus rhododactylus.*

**Table 1 insects-11-00856-t001:** Known localities of *Phloeotribus rhododactylus* in Europe. Data and localities are summarized based on the literature available from Google Scholar, Zobodat, Biodiversity Heritage Library, and www.gbif.org.

Country	Localities (County) and References
Austria	Karnburg [46]; Donnerskirchen, Eisenstadt, Gars, Gemeinlebarn, Horn, Stein an der Donau, Winden, Yois [47]; Niederdonau [48]; Helenental, Kalksburg [49]; Mitterberg [50]
Belgium	Antwerpen, Baileux, Borsbeek, Breuvanne, Cerfontaine, Cielle, Cour-sur-Heure, Dourbes, Eupen, Grandglise, Furnaux, Hantes-Wihéries, Hermeton-Sur-Meuse, Henripont, Jamiolle, Jamoigne, Lambermont, Marchienne-Au-Pont, Marloie, Meeuwen, Mont-Gauthier, Neeroeteren, Neufmoulin, Nivelet, Odrimont, Oignies-en-Thiérache, Rachecourt, Robelmont, Roelen, Samrée, Sosoye, Vodecée, Wauthier-Braine, Wilsele-Dorp, Zeveneken [51]
Bulgaria	Kaspichan [51]
Croatia	Draga, Krapina, Križevci, Orehovica, Sljeme, Trakoščan, Zagreb [52]
Denmark	Allinggård, Funder, Nørholm, Vejle [53]; Fuglsǿ [51]
France	Albaran, Avignon, Bassin de la Seine, Bois du Rouvray, Brout-Vernet, Corse, Env. de Rodez, Grocy, Hyères, Iles de la Loire, Isdes, Marcilly-en-Vilette, Marseille, Mers, Montfaucon, Montpellier, Mt. Ventoux, Murles, Peille, Sle.Maxime, Vosges [54]; Abzac, Belin-Béliet, Lège-Cap-Ferret, Le Verdon-Sur-Mer, Queyrac, Quinsac, Saucats [55]; Anost, Arandon, Archail, Arnières-sur-Iton, Beaumont du Ventoux, Bionville, Blagon, Bonnée, Brantes, Braux, Chambry, Champagne-sur-Seine, Champsecret, Chapeau, Chemilly, Cheval-Blanc, Cruzille, Eckartswiller, Eschbourg, Fontbelle, Héches, Henrichemont, La Borne, La Javie, La Môle, Lavault-de-Frétoy, Le Bourgneuf, L’Épine, Le Vallée Heureuse, Les Bréviaires, Les Choux, Les Ferrands, Les Mayons, Le Poët-Célard, Le Valtin, Maisse, Ménerbes, Méolans-Revel, Mijanés, Milly-la-Forêt, Mimizan, Monferran-Savès, Mongausy, Mons, Montigny-lès-Cormeilles, Mozac, Neuvy-sur-Barangeon, Nohédes, Orcemont, Ouzouer-sur-Loire, Raon-sur-Plaine, Revalies, Rotis, Sagy-le-Bas, Saint-Béat, Sainte-Geneviève-des-Bois, Saint Estéve, Sainte-Tulle, Saint-Floren, Saint-Lys, Saint-Pierre-du-Lorouër, Saintry-sur-Seine, Salerm, Sare, Saurat, Savigny-le-Temple, Sermoyer, Sotteville-lès-Rouen, Tartonne, Vergons, Vineuil-Saint-Firmin [51]
Germany	Nördlinger [56]; Bokelsberg, Göhrde [57]; Hamburg, Mecklenburg, Oldenburg, Pommern, Sachsen, Thüringen, Westfalen [58]; Baden, Bayern, Hessen, Nassau, Wurtemberg [59]; Lindenberg, Sasbachwalden [60]; Bad Herrenalb [61]; Freiburg in Breisgau [62]; Blankenburg, Halberstadt [63]; Bluno, Döberitz, Terra Nova [64]; Boostedt, Eilsdorf, Flittard, Heenes, Kleinraschütz, Nettekoven, Niederhausen, Pfeifenkrug [51]
Greece	Zachlorou [65]
Hungary	Budapest [66]; Szöce [67]; Vas [68]
Ireland	Cork, Dublin, Kerry, Wexford, Wicklow [69]
Italy	Vallombrosa [70]; Sardegna [71]; Sicilia [72]; Nuoro, Porta [73]; Fennhals, Oberfennberg [50]; Montegiovi [51]
Luxembourg	Goebelsmühle, Nordwestl. Altrier, N. Troisvierges, Südwestlich Berchem [74]; Hoscheid, Wahlhausen [75]; Luxembourg, Michelau, Rodenbourg, Sassel [51]
Netherlands	Huizen [76]; Amerongen, Baarn, Partij [51]
Poland	Gorzów Wielkopolski [77]; Pomorz Zachodni [78]; Brójce, Glińsk, Ołobok, Przygubiel, Rzeczyca [79]; Eastern Sudets Mts., Lower Silesia, Polanow, Trzebnica Hills, Western Beskid Mts. [80]; Kędrzyno [81]
Portugal	Pelados [82]
Romania	Domogled [66]
Russia—Kaliningrad	Gurevsky district [83]
Slovakia	Slovenský Búr [15]; Mlýňany [19]
Spain	Galicia [17]; Andorra, Barcelona, Valle de Arán [84]; País Vasco [85]; Salamanca [86]
Sweden	Norje, Vanneberga [51]
Switzerland	Mittelland [87]; Arbedo, Capolago, Caviano, Chiasso, Lavigny, Loco, Lugano, Morges [51]
Turkey	Antalya [88]
United Kingdom	Bewdley Forest, Birch Wood, Chatham, Coombe Wood, Darenth, Dartford, Durham, Eastbourne, Forth, Herefordshire, Liverpool, Mickleham, Monmouthshire, Moray, New Forest, Northumberland, Reigate, Rusper, Scarborough, Sheerness, Shirley, Shirley Warren, Southampton, Southend, Southsea, Tay, Tweed, West Wickham, Whitstable, Wimbledon, Woking [89]; Aberdeen [90]; Ainsdale-on-Sea, Aldridge, Annan, Ashford, Ashtead, Aycliff, Aylesford, Barston, Bermuda, Bewbush, Binley Woods, Bishop’s Castle, Blackborough End, Blythburg, Botcherby, Brancaster, Bredgar, Castle Rising, Charleston, Chithurst, Claydon, Coventry, Dale, Derwen Fawr, Dinas Dinlle, Donnigton, Dordon, Downside, Dungeness, Eyke, Eythorne, Ferndown, Foel, Frodesley, Garboldisham, Gladestry, Glan-yr-afon, Gooderstone, Great Hockham, Gretna, Great Wenham, Gumley, Gwbert, Hales Place, Hartfield, Hornton, Hoylake, Kidderminster, Kingston upon Thames, Knockin, Lakenheath, Langwathby, Llangaffo, Longfield, Maltby, Marholm, Moriah, Muddles Green, Mundford, Nantmel, Narborough, Newcastle, Newton, Newton St. Faith, North Wootton, Orford, Pennparcau, Pennrhos Garnedd, Petworth, Port Talbot, Priors Hardwick, Red Lodge, Richings Park, Riddlesworth, Ruthin, Ruyton-XI-Towns, Seale, Sea Palling, Selattyn, Stratford-upon-Avon, Swansea, Tankersley, Tattingstone, Thetford, Ullenhall, Undley, Upper Whiston, Walsall, Walton, Washington, Wealden, West Bay, Whitwell, Wimborne Minster, Winchelsea, Wolverhampton, Woolston, Wootton [51]

**Table 2 insects-11-00856-t002:** Known localities of *Phloeotribus rhododactylus* in the Czech Republic. Data were excerpted from the literature, as well from private collections and from unpublished entomological reports (NM—Natural Monument; NR—Natural Reserve). Collection of data by T. Fiala in 2020 is described in this report.

Locality	Coordinates	Altitude (m a.s.l.)	Year of Collection	Source
**Český Těšín**	49.7418600 N, 18.5915097 E	340	?	[19]
**Domažlice**	49.4605044 N, 12.9259797 E	500	?	[18]
**Dvorek**	50.1351472 N, 12.4058650 E	450	2020	T. Fiala
**Havraníky**	48.8170047 N, 16.0035081 E	330	2018	J. Vávra, J. Procházka
**Havraníky**	48.8190236 N, 16.0027867 E	340	2004	[23]
**Cheb**	50.0771789 N, 12.3266058 E	500	?	[19]
**Lázně Kynžvart**	50.0077714 N, 12.5948347 E	580	2020	T. Fiala
**Nýdek (Velká Čantoryje Mt.)**	49.6712786 N, 18.7798358 E	995	?	[15]
**Pístov**	49.9155872 N, 12.7591886 E	530	2020	T. Fiala
**Popice**	48.8263717 N, 16.0041411 E	350	2013	O. Konvička
**Popice**	48.8196681 N, 16.0115225 E	290	2003	O. Konvička
**Popice**	48.8255664 N, 16.0081214 E	320	2000	R. Szopa
**Popice**	48.8209906 N, 16.0081136 E	300	1986	[95]
**Popice**	48.8248178 N, 16.0127778 E	300	2019	J. Vávra, J. Procházka
**Pouzdřany**	48.9494600 N, 16.6356497 E	250	?	[96]
**NM Oleksovické vřesoviště**	48.8969431 N, 16.2476167 E	240	2017	[97]
**NR Údolí Oslavy a Chvojnice**	49.1415208 N, 16.2193547 E	320	2013	[98]
**Telnice**	50.7241322 N, 13.9681581 E	330	1977	V. Týr
**Teplička**	50.1532528 N, 12.8491658 E	460	2020	T. Fiala

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
