# Peer review of "The Bark Beetle Phloeotribus rhododactylus (Coleoptera: Curculionidae) Has a Stable Range in Europe"

_insects, 2020, doi:10.3390/insects11120856_

Round 1

Reviewer 1 Report

The subject matter in the article and the basic approach is very interesting.  There are too many articles about bark beetles in conifers and not enough about the really understudied species such as Phloeotribus rhododactylus.  I enjoyed reading it.  Nonetheless the article requires major revision in its current form.

  1. The authors state repeatedly that this is a Mediterranean species.  It’s hard to condsider a species that is apparently abundant in the British Isles to be “Mediterranean”. In addition to the European localities they neglect to cite Algeria, Egypt, Libya, Morocco, and Tunisia (Wood, S.L.; Bright, D.E. 1992. A catalog of Scolytidae and Platypodidae (Coleoptera), Part 2.  Taxonomic Index (Volumes A,B). Great Basin Naturalist Memoirs. 13: 1-1553, page 228 : http://www.barkbeetles.info/wood_bright_92_catalog_js.php?lookUp=228).
  2. Based on the hosts listed and countries listed by Wood & Bright I would assume that the beetle is found throughout Spain, Italy and Greece, probably also in Syria, Lebanon, and Israel.. I believe that the authors are considering absence of evidence to be evidence of absence.  That is lack of records means that the species is absent from certain areas.  The United Kingdom and France have been heavily sampled.  I suspect that the lack of collection records in Spain, Italy and Greece is purely due to lack of sampling, not absence of the beetle. 
  3. They refer to the host plant, Cytisus scoparius as being native to central and western Europe, but also mention that it is considered invasive in eastern Europe, and presumably northern Europe (all the way to Scotland and Sweden). There is no evidence provided that either the host plant or beetle have extended their ranges northwards and eastwards.  While the density of points seems to diminish eastwards of the Czech Republic, it is not clear if this is “real” or due to lack of sampling.
  4. Is scoparius present around the Mediterranean basin in southern Spain, Italy, Greece and northern Africa,
  5. There is also no evidence given that collections on the eastern or northern edges of the species range are more recent than western collections that would indicate recent spread (i.e., within last 100 years).
  6. The authors mention that low temperatures might be a factor. It would help considerably if they could superimpose isotherm lines on the distribution map.
  7. Any maps showing change in distribution of the host plant would be useful.
  8. I consider that the presence / absence of rhododactylus in the eastern Czech Republic to be significant because the authors clearly show in their map (inset C) that presence and absence are related to elevation (and associated temperature differences).
  9. In the end, whether or not the beetle has a stable range in eastern Europe has not been demonstrated. GBIF data are valuable, but I believe that the authors are using them uncritically.  I suspect that the authors are probably correct, but they need to organize their evidence better,

Author Response

1. The authors state repeatedly that this is a Mediterranean species. It’s hard to condsider a species that is apparently abundant in the British Isles to be “Mediterranean”. In addition to the European localities they neglect to cite Algeria, Egypt, Libya, Morocco, and Tunisia (Wood, S.L.; Bright, D.E. 1992. A catalog of Scolytidae and Platypodidae (Coleoptera), Part 2. Taxonomic Index (Volumes A,B). Great Basin Naturalist Memoirs. 13: 1-1553, page 228 : http://www.barkbeetles.info/wood_bright_92_catalog_js.php?lookUp=228).
Authors: We wrote only one place (eg. Line 149) that this species is Euromediterranean, but reviewer has right that we should other non-european countries, added in line 149-150.
2. Based on the hosts listed and countries listed by Wood & Bright I would assume that the beetle is found throughout Spain, Italy and Greece, probably also in Syria, Lebanon, and Israel.. I believe that the authors are considering absence of evidence to be evidence of absence. That is lack of records means that the species is absent from certain areas. The United Kingdom and France have been heavily sampled. I suspect that the lack of collection records in Spain, Italy and Greece is purely due to lack of sampling, not absence of the beetle.
Authors: Yes, this is right we added in line 144-145.
3. They refer to the host plant, Cytisus scoparius as being native to central and western Europe, but also mention that it is considered invasive in eastern Europe, and presumably northern Europe (all the way to Scotland and Sweden). There is no evidence provided that either the host plant or beetle have extended their ranges northwards and eastwards. While the density of points seems to diminish eastwards of the Czech Republic, it is not clear if this is “real” or due to lack of sampling.
Authors: We do not understand, we did not write anywhere about invasion in Europe. We agree that there is no evidence about extension of range both, host as well as bark beetle. Occurrence in the Czech Republic is widely commneted in the chapter „Occurrence of Phloeotribus rhododactylus on eastern edge of its range“. Even in areas with abundant occurrence of Cytisus scoparius, Ph. rhododactylus is rare.
4. Is scoparius present around the Mediterranean basin in southern Spain, Italy, Greece and northern Africa,
Authors: Yes, added in line 95
5. There is also no evidence given that collections on the eastern or northern edges of the species range are more recent than western collections that would indicate recent spread (i.e., within last 100 years).
Authors: We do not think that the species would spread, on the contrary, we try to prove that the area is stable. We have added in lines 112-118 the whole paragraph that characterised distribution up to first half of 20th century and also did new map (Fig. 2B).
6. The authors mention that low temperatures might be a factor. It would help considerably if they could superimpose isotherm lines on the distribution map.
Authors: We have added annual mean temperatures into map (Figure 2A).
7. Any maps showing change in distribution of the host plant would be useful.
Authors: There was no change in distribution of the host plant so creating such a map is impossible.
8. I consider that the presence / absence of rhododactylus in the eastern Czech Republic to be significant because the authors clearly show in their map (inset C) that presence and absence are related to elevation (and associated temperature differences).
Authors: It is true only in the case of locality Nýdek. The second one ÄŚeský TÄ›šín lies at 300 m asl. We suggestes that the main reason is small places of C. scoparius in long distance to each others. Added in line 174-176.
9. In the end, whether or not the beetle has a stable range in eastern Europe has not been demonstrated. GBIF data are valuable, but I believe that the authors are using them uncritically. I suspect that the authors are probably correct, but they need to organize their evidence better,
Authors: We think that when we add evidence of the expansion of the barkbeetle in the 19th century and in the first half of the 20th century, we demonstrate the stability of the area. Also the situation in the Czech Republic proves that the incidence is the same.

Reviewer 2 Report

The note is good and essentially correct, just a minor changes seems necessary, see the few notes in the attached file.

Author Response

Line 13: complicatus – corrected line 13
Line 40: Not all. Better to change as follows: "The aboslute majority of Scolytinae beetles..." – corrected line 40
Line 57 and 58: "Cytisus scoparius" goes in italic and "P. rhododactylus" goes in italic
- corrected lines 57-58
Line 94: unnecessary, eliminate words highlighted in yellow – accepted line 94
Line 105: comma, not parethesis – corrected line 105
Line 135: Euromediterranean – corrected line 149
Line 140: already stated. Eliminate the highlighted sentence – accepted line 155
Line 163: Eliminate this meaningless sentence – deleted in line 158
Line 189: add "species" after rare – added line 208

Reviewer 3 Report

Comment
The authors report a brief review of the data on the biology of Phloeotribus rhododactylus, clearly privileging however the data on its distribution in Europe, mainly based on data found and already reported in detail online (see GBIF reference 49). Currently, the available geographical data of this widespread Western Palaearctic species are especially numerous in some countries such as UK, Belgium and France than in other countries. The authors show that this species is also widely distributed in the Czech Republic by personal collection and the study of collections of other Czech colleagues. It is probable that the study of collections from e.g. Switzerland, Italy, Germany, Slovakia, Hungary might give similar results and it seems that the lack of more numerous reports in some counties is only due to the lack of more careful researches on the widely distributed main host plant Cytisus scoparius.

I suggest that this paper might be presented as article, markedly stressing the new data of distribution of this species in the Czech Republic and therefore its probable more substantial presence in Eastern countries.

Author Response

I suggest that this paper might be presented as article, markedly stressing the new data of distribution of this species in the Czech Republic and therefore its probable more substantial presence in Eastern countries.

Response – added „the Czech Republic“ into keywords and change of goals in line 60 to be more specific.

Round 2

Reviewer 1 Report

The authors were able to address my earlier concerns with some relatively minor changes and additions that better documented the full range of the beetle and the historical basis for their observations about the spread of the host plant versus spread of the beetle.  The references to Pfeffer were particularly valuable.

This is a good and interesting paper and I hope to see more on middle European bark beetles from the authors

Author Response

No reply for the reviewers 1, because he has no comments

Reviewer 3 Report

I agree with the changes made by the authors

Author Response

No reply for the reviewer 3, because he has no comments